# Surface-Treated Recycling Fibers from Wind Turbine Blades as Reinforcement for Waste Phosphogypsum

**DOI:** 10.3390/molecules27248668

**Published:** 2022-12-08

**Authors:** Lilin Yang, Weilin Zhao, Daobei Wang, Yang Liu, Dongzhi Wang, Na Cui

**Affiliations:** 1School of Materials Science and Engineering, University of Jinan, Jinan 250000, China; 2School of Civil Engineering and Architecture, University of Jinan, Jinan 250000, China

**Keywords:** waste fiber, wind turbine blade, phosphogypsum, surface treatment

## Abstract

An attempt at the treatment of the waste fiber (WF) from the wind turbine blade (WTB) was made through the modifier of dopamine hydrochloride and the compound modifier of dopamine hydrochloride and 2,5-dihydroxy terephthalic acid or 3,4-dihydroxy cinnamic acid or 3,4-dihydroxy benzonitrile, corresponding to obtain four modified waste fibers (MWF1, MWF2, MWF3, and MWF4). The MWFs samples’ microstructure properties were characterized using SEM, EDS, XPS, FTIR analyses, and water contact angle tests. The results revealed that all the MWF surfaces were wrapped by a distinct coating layer and had different elemental compositions and chemical groups, demonstrating the significant effect of the four modifications on the WF surfaces. The hydroxyl, amino, or nitrile groups were grafted onto the WF surfaces causing improvement of the hydrophilicity and reactivity. Furthermore, all the MWFs as the reinforced materials were incorporated into the industrial waste phosphogypsum (PG) to manufacture the phosphorous-building gypsum composites (PBGC). The effects on the micro-morphology and mechanical properties of the PBGC were evaluated. The results also show the improvement in flexural and compressive strength with the addition of MWFs into the PBGC, due to the enhancement of the compactness between the MWF and phosphogypsum matrix. In particular, the effects of three compound modifiers on the flexural and compressive strength are more significant. The highest flexural and compressive strength was contributed by the PBGC-MWF4 with 2% dosage using a compound modifier of dopamine hydrochloride and 3,4-dihydroxy benzonitrile, which were enhanced 61.04% and 25.97% compared with the PBG.

## 1. Introduction

Human society has been facing the challenge of the energy crisis due to factors such as population growth, petroleum scarcity, excess carbon dioxide emissions, global warming, and new technical emergence [1]. Wind energy, as a clean, renewable energy resource [2], has become one of the promising ways to solve the energy problem. The generation of wind energy is a process that utilizes the specially designed turbine blade to absorb the kinetic energy of wind and converts it to mechanical energy, and further drives the generator rotating to obtain electric power [3]. With the growth in the use of wind energy, the demand for wind turbine blades (WTB) as one of the most important components keeps increasing sharply [4]. Generally, the WTB is essentially composed of a metal structure enveloped in epoxy resin and glass fiber, combined with wood and a curing agent. Their average life cycles are generally 20 years [5]. Recently, wind turbine blade composites contained glass fiber and epoxy. In the wind turbine blade manufacturing process, several types of high-strength fibers, such as carbon fibers, basalt, aramid fibers, S-glass, R-glass, and E-glass fibers have been used as reinforcement agents. However, considering the cost factor, in China, wind turbine blades usually use E-glass fiber as the reinforced materials for economic and environmental reasons. Once the WTB reaches the end of the service life, it will be disposed of as waste [6]. It was reported that each abandoned WTB could generate around 12 tons of waste [7,8]. Consequently, it is essential to develop some methods to deal with the waste of the WTB. Currently, the most uncomplicated process for getting rid of this waste is to pile them up or in massive landfills. However, this is not an economically and environmentally friendly process. The best route is recycling [9], repurposing [10,11], and repairing [12]. Nevertheless, the exploration process is challenging and still requires further deep investigation. Oliveira et al. [13] have applied the waste collected from the WTB cutting process to some civil constructions and demonstrated the feasibility of incorporating this solid waste into cementitious materials. Baturkin et al. [14] also tested the effects of turning WTB waste glass fiber-reinforced polymer materials into concrete. The results showed that the mechanical properties are comparable to normal concrete materials.

PG is another industrial waste when producing phosphoric acid and phosphate fertilizer [15,16]. At present, about 200 million tons of PG is generated annually [17]. This waste also poses severe environmental problems. An effective solution is that PG is pretreated and calcined to be prepared in all kinds of phosphorus-building gypsum (PBG) products. Therefore, recycling and reusing PBG along with WTB waste fibers is an effective way to produce high-performance “green” building materials [18,19]. Although after proper treatment, PBG can be used to manufacture products, such as paper gypsum board [20], porous sound-absorbing material [21], building standard brick [22,23], and as gypsum cementitious materials [24], the mechanical properties and durability are still not able to satisfy the requirements of the fields.

To improve physical and mechanical properties, reinforcements such as natural and synthetic fibers were combined with the matrix to enhance the cracking resistance, strength, and toughness of composites [25,26]. Hua [27] found that mixing glass fibers with phosphogypsum can improve its overall performance. Similar results stated that adding recycled glass fibers into a gypsum matrix could enhance flexural strength [28]. Although glass fibers have been approved as an effective agent to reinforce phosphogypsum, it is still challenging to obtain high-performance cementitious materials by using solid waste as raw materials to realize cost-effective and environmentally friendly manufacturing. Without a proper treatment method, the reinforcement effect will be significantly limited due to the weak interfacial bonding between the recycled fibers and the matrix of the cementitious material.

It was found that surface treatment is an effective way to enhance the interfacial bonding between the fibers and the matrix in composites [29]. Unlike previous studies, in which new glass fiber was mixed with the gypsum matrix as the reinforcement agent, the mechanical properties of the waste fiber/gypsum composites declined sharply due to the poor bonding strength between the waste glass fibers and the gypsum matrix. As a result, it is reasonable to explore proper surface treatment methods to realize the effective enhancement of properties in recycled fiber-reinforced phosphogypsum composites. In this study, waste glass fibers obtained from WTB were used as reinforcement agents to manufacture high-performance phosphogypsum-based cementitious materials. Dopamine hydrochloride (dopamine), dopamine compound with 2,5-dihydroxy terephthalic acid (terephthalic), 3,4-dihydroxy cinnamic acid (cinnamic), and 3,4-dihydroxy benzonitrile (benzonitrile) were utilized as a surface treatment agent to modify the surface condition of the waste glass fibers and thus improve the adhesion between the waste glass fibers and the phosphogypsum matrix. Dopamine is a biological neurotransmitter with the advantage of spontaneously polymerizing to form polydopamine. Polydopamine has a function that could firmly adhere to various substrates, including wood, plastics, and metals [30,31]. The terephthalic, cinnamic, and benzonitrile are highly polarized compounds as intermediates in an organic synthesis process. They have similar molecular structures of catechol and dopamine and excellent compatibility with the same high polarity of dopamine material. Hence, these four modifiers were selected to modify the WF to form a better interaction between the recycled waste glass fiber and the phosphogypsum matrix. The objective of this study is not only to provide an effective and cost-effective way to recycle solid wastes but also to shed light on how to change the surface condition of reinforcement elements to optimize the final properties of solid waste-based cementitious materials.

## 2. Experimental Procedure

### 2.1. Materials

The WTB waste glass fibers were obtained from Jiuquan Wind Power Base (Gansu, China). Dopamine, terephthalic, cinnamic, benzonitrile, and tris-hydrochloride were purchased from Aladdin Biochemical Technology Co., Ltd. (Shanghai, China). All chemicals were analytically pure. The phosphogypsum was provided by Shandong Zhengwei Agricultural Technology Co., Ltd. (Jinan, China). The superplasticizer was purchased from Shandong Jianke Building Materials Co., Ltd. (Jinan, China), and the retarder was tartaric acid from Kemiou Chemical Reagent Co., Ltd. (Tianjin, China).

### 2.2. The Waste Glass Fibers

The waste glass fibers were obtained in two steps, as shown in Figure 1. The first step was processed at Jiuquan Wind Power Base. In this step, the abandoned WTB were cut into blocks using a diamond wire saw, which cut peer blocks of about 1 m^2^. Then, the blocks were shredded, crushed by a crusher, and the fibrous material was pulverized with a grinding mill. (The factory specially customized the machine of the crusher and grinding mill. The equipment had four parts: shredding part, transmission equipment, primary crushing equipment, and vibrating screen.). The high-performance multi-functional grinding mill (XICHU-4500Y, Zhejiang, China) ground the sheets into fibrous materials. The fibrous materials were agglomerated like cotton. The second step was sieving, washing with deionized water to remove the floc, and drying the fibers in an oven at 80 ℃. The final products were the WFs with a length of about 4–9 mm. The length–diameter ratio was not unique.

### 2.3. Surface Treatment of the WFs

Four types of surface modifiers, including dopamine, a mixture of dopamine and terephthalic, dopamine and cinnamic, and dopamine and benzonitrile were used to change the surface conditions of the waste glass fibers. The waste glass fibers modified by these four modifiers were named MWF1, MWF2, MWF3, and MWF4, respectively, and the unmodified waste glass fiber was named UWF.

Figure 2 presents the schematic demonstration of the surface treatment process. As illustrated in this figure, first, 25 g of UWFs were immersed in 250 mL Tris-HCl buffer solution with pH = 8.5 and stirred for 20 min, with a stirring rate of 300 r/min, to obtain the UWF suspension at room temperature [32]. Subsequently, 0.5 g modifiers were divided into five parts and added to the above suspension every five minutes, and a uniform solution was obtained. At that moment, the UWF surface had gradually changed from white to gray/brown. Then, the waste fibers were soaked in the stirring solution at room temperature for 24 h, followed by deionized water washing to neutralize the fibers. Finally, the products were collected by filtration and dried at 80 °C for 2 h to obtain the MWF1.

The other three modification steps were similar to the MWF1, which used 0.25 g dopamine with 0.25 g terephthalic, cinnamic, or benzonitrile, respectively, to react for 24 h at room temperature to acquire the MWF2, MWF3, and MWF4. Figure 2 presents the schematic demonstration of the surface treatment process.

### 2.4. Characterization of the Waste Glass Fibers

The morphology and elemental analysis were characterized by a scanning electron microscope (Zeiss Sigma 300) with an energy spectrum (EDX). The structural properties were described by Fourier transform infrared (FT-IR) spectra (Nicolet 380, Thermo, American) and X-ray photoelectron spectroscopy (XPS) (AXIS SUPRA, Jin Island, Japan). The hydrophobic performance was evaluated by a water contact angle measuring instrument (JC2000D1, Shanghai, China)

In this study, with a phosphogypsum to water ratio of 1:2.5, the phosphogypsum was washed to neutral, and the organic pollutants, soluble phosphorus, fluorine, and alkali were removed by this washing process. After, the phosphogypsum was calcined in an oven at 180 °C for 3 h, ground with a ball milling procedure, followed by 15 days of natural drying to obtain the PBG. Table 1 lists the composition of the PBG by the X-ray fluorescence (Tiger S8, BRUKER, Germany). The waste glass fiber modified with PBG was produced with a water-to-binder ratio of 0.65 and named PBGC-UWF, PBGC-MWF1, PBGC-MWF2, PBGC-MWF3, and PBGC-MWF4. The PBGC without waste glass fibers was named PBGC-NWF. The length of the new glass fiber was about 9 mm, similar to the WF. The PBGC with new glass fibers was named PBGC-NGF. The dosages of retarder and superplasticizer were fixed at 0.02% and 0.9%, respectively. The retarder was tartaric acid, and the superplasticizer was a polycarboxylic water reducer. The amounts of the UWF or MWF or NGF were 0, 5, 10, 15, 20, and 25 in 1000 g PBG, respectively. The mixture proportions are listed in Table 2. Figure 3 depicts the sample preparation process. The dry UWF or MWF, PBG, and retarder were mixed in a plaster mixer with a low speed of 300 rpm for 9 min and then stirred at a low speed of 300 rpm with the addition of water and superplasticizer for 1 min followed by a high mixing speed of 800 rpm for 2 min. Fresh samples were cast into molds with a size of 40 mm × 40 mm × 160 mm and cured for 2 h at room temperature. Finally, the examples were demolded to obtain the PBGC sample. After demolding, the PBGC samples were cured for another 7 days with a relative humidity of 50 ± 5% at 20 ± 2 °C. After 7 days of curing, the samples were dried in an oven at 40 ± 4 °C until reached a constant weight.

### 2.5. Properties Analysis of the PBGC

The mechanical properties were tested by an integrated compressive and flexural machine (SANS, China) according to the Chinese standard requirements [32]. The loading rate of the flexural and compressive strengths was 50 N/s and 2 mm/min. The flexural strength was tested by using three 40 mm × 40 mm × 160 mm specimens, and each of the half specimens after flexural strength testing was used as the compressive strength testing. The final results were the average value of three samples. The pore structures were evaluated by mercury intrusion porosimetry (MIP) (AutoporeV9620, American). The low and high pressures of the measurement were controlled at 0.003 MPa and 410 Mpa, respectively. Scanning electron microscope (SEM), EVO/LS15, provided by Germany Zeiss SMT Co., Ltd., Oberkochen, Germany, was used to investigate the morphology of the surface of the waste glass fibers and the fracture surface of the composites. The XRD pattern was obtained on a Rigaku D/max-rA X-ray diffractometer with Cu Kα radiation (λ = 1.5406 Å) at 45 kV and 20 mA. The diffraction scans were performed from 5° to 90° at a rate of 1°/min, with a step size of 0.02°.

## 3. Results and Discussion

### 3.1. Characterization of All the WFs

The SEM/EDS morphologies of the modified waste glass fibers (MWF1, MWF2, MWF3, MWF4) are shown in Figure 4. As can be seen from this figure, the surface of the UMF is distinctively smooth and changed to rough conditions in all MWFs. It can be claimed that the surface treatment methods have successfully changed the surface conditions of the waste glass fibers.

Figure 4 presents the SEM surface images of the waste glass fibers and the corresponding EDS analysis of a selected point. It could be observed that C, O, Si, and some metal elements on the UWF and all the MWF surfaces are the main elements on the surface of the waste glass fibers. It could also be observed that the amount of Si was reduced after the surface treatment. In particular, when the compound modifier was covered to the surface of MWF2, MWF3, and MWF4, the Si amount dropped from 11.31% to 3.83%, 3.59%, and 4.45%, indicating the feasibility of partial replacement of terephthalic, cinnamic, and benzonitrile. Besides, unlike the UWF, all the MWF surfaces still contained a small amount of N. The amount of N (13.15%) on the surface of MWF4 is attributed to the double effects of amino from dopamine and nitrile group from benzonitrile.

The presence of coating films on the MWF surfaces was further confirmed by XPS analysis. The full spectrum and zoomed spectrum of the UWF and four MWFs are illustrated in Figure 5. In Figure 5a–e, the XPS results demonstrated that the UWF and four MWFs were primarily composed of C1s, N1s, O1s, and Si2p elements. The zoomed spectrum in Figure 5 revealed that the surface conditions of the waste glass fibers before and after surface treatment significantly depended on the different chemical states of elements C and N in deconvoluted core-level spectra because Figure 5(a-3–e-3) presented the same O1s XPS spectrum and Figure 5(a-4–e-4) showed spectra identical Si2p XPS spectrum [33]. The spectrum of C1s in the UWF showed two fitting peaks at 285.6 eV and 284.1 eV respectively are originated from the C-O-C and C-C/C-H (shown in Figure 5(a-1)). The peaks at 398.6 eV, 399.7 eV, and 400.5 eV of N1 spectra correspond to the bonds of C-N-C, N-C, and N-H, respectively (Figure 5(a-2)). After being decorated by the modifiers, compared with the spectra of UWF and C1s spectra of MWF1, MWF2 and MWF3 appeared a new peak of C=O in Figure 5(b-1,c-1,e-1). In contrast, some peaks of the N1s spectra, including two peaks of N-C and N-H groups in Figure 5(b-2,e-2) and a peak of N-C group in Figure 5(c-2,d-2), have disappeared. In comparison with the element content of UWF, it could be found that the C1s content of MWF1 increased due to the modification of dopamine. In contrast, the C1s content of MWF2, MWF3, and MWF4 was slightly lower than that of the MWF1 due to the appearance of terephthalic, cinnamic, and benzonitrile instead of partial dopamine, while the N1s content of MWF1, MWF2, MWF3, and MWF4 increased because of N-C, N-H, and C-N-C groups. It was clear that the modifiers were successfully grafted onto the surface of the waste glass fibers and formed an adhesive coating.

To further investigate the chemical structure vibrations of the waste glass fibers before and after the surface treatment, the FTIR spectra of all the waste glass fibers were tested, and the results are shown in Figure 6. In the FTIR spectra of the UWF, it can be found that the distinct peak at 3400 cm^−1^ was ascribed to the stretching vibrations of O-H and N-H presented in resin. The peak at 2961 cm^−1^ corresponded to the aliphatic C-H stretching, and the peaks at 1765 cm^−1^, 1240 cm^−1^ (aromatic C=O), and 845 cm^−1^ (epoxy group Antisymmetric Vibration) presented the thermoset epoxy group. The peaks at 845 cm^−1^ and 770 cm^−1^ evidenced the Si-O band and the presence of glass fiber. This abundant evidence proves WF contained glass fiber and epoxy resin. The peak at 1502 cm^−1^ was assigned to the aromatic C=O from the curing agent. After modification, the band intensity at 3400 cm^−1^ shows a clear distinction with the change of the types of the modifier due to the increase of -NH- and -OH groups. Meanwhile, a few new peaks, such as the peaks at 1647 cm^−1^ and 1250 cm^−1^, which originated from the bond of C=O and C-O, can be observed [34]. The typical peaks at 1714 cm^−1^ corresponding to the dimeric carboxyl in terephthalic acid (MWF2), cinnamic acid (MWF3), and cyanophenyl (MWF4) demonstrated the band of C=O stretching vibration. In particular, the C≡N stretching vibration band at 2300 cm^−1^ can be observed in the spectrum of the MWF4. These changes were due to a modifier’s introduction on the surface of the glass fibers.

Figure 7 gives the impact of the surface treatment on the contact angle of the UWF and all the MWFs. It can be seen that the initial contact angle of the waste glass fiber was 103.76°, which confirmed the hydrophobic innate of the surface. Even after being soaked for 60 s, the contact angle only slightly dropped from 103.76° to 93.87° due to the lack of hydrophilic groups on the surface. Conversely, the initial contact angles of MWF1, MWF2, MWF3, and MWF4 were successively reduced to 68.53°, 49.75°, 45.53°, and 0°. This revealed that the UWF was successfully transformed from hydrophobic to hydrophilic after treatment by the modifiers. In other words, it implied that the coating layer of the modifier was grafted onto the surface of the waste glass fibers. Additionally, the best hydrophilicity exhibited on the MWF4 surface was modified by dopamine and benzonitrile because the amount of hydroxyl, amino, and nitrile groups were much higher than the others.

Figure 8 illustrates the schematic description of the surface modification process. Dopamine can be oxidized by being dissolved oxygen in an alkaline buffer aqueous solution and spontaneously self-polymerized to form a polydopamine film on the WF surface. The hydroxyl (-OH) and amino (-NH2-) groups in the molecular structure of dopamine were directly grafted on the surface of the waste glass fibers, as illustrated in Figure 8a.

The chemical structure of terephthalic, cinnamic, and benzonitrile contains carboxyl (-COOH-) and nitrile (-CN-) groups with hydrophilicity. When they appeared in the dopamine polymerization system, the carboxyl groups in terephthalic, cinnamic, or nitrile groups in benzonitrile were grafted on the polydopamine film of the surface of the waste glass fibers, as shown in Figure 8b–d. The small molecules were embedded into polymer voids through hydrogen bonds and formed the synergistic polymer film, which was more condense than dopamine. The benzonitrile molecules were relatively regular and easy to arrange in the polymer film [35,36,37,38]. Compared with the aromatic acids of terephthalic and cinnamic, it was easier to form orderly and regular high-quality films on the surface of the material, leading to superior hydrophilicity.

### 3.2. Microstructure and Mechanical Properties of the PBGCs

The SEM images of the fracture surfaces of the PBGC samples with a waste glass fibers dosage of 2.0% are displayed in Figure 9a–f. It could be found that the PBG was needle-like lamellar crystals with non-universal sizes. The PBGC-NWF had holes and cracks (Figure 9a). When the waste glass fibers were randomly distributed in the PBG (Figure 9b–f), the gypsum holes and cracks were filled. Meanwhile, the waste glass fibers in the PBG matrix can serve as bridges to promote the continuity of the PBG matrix.

To evaluate the variation of PBGC voids, the porosity of all the PBGCs was conducted by the MIP. The results are illustrated in Figure 10. It can be seen that the pore size of all the tested samples ranged from 0.5 μm to 5 μm. If the dominant peak of the PBGC-NWF with the median pore diameter of about 1.82 μm was taken as a target, an interesting phenomenon can be found that the dominant peak of PBGC-UWF shifted to the smaller pores (1.65 μm), the location of the dominant peak of PBGC-MWF1 m modified by dopamine (at about 1.8 μm) was approximately in agreement with that of the PBGC, while those treated by a compound modifier (PBGC-MWF2, PBGC-MWF3, and PBGC-MWF4) moved to a larger pore size that was more than 2 μm. Nevertheless, it can be seen in Table 3, the cumulative pore volume of all the PBGCs gradually falls based on this order of the PBGC-NWF, PBGC-UWF, PBGC-MWF1, PBGC-MWF2, PBGC-MWF3, and PBGC-MWF4. By contrast, the porosity of UWF, MWF1, MWF2, MWF3, and MWF4 gradually decreased, and were 11.99%, 14.73%, 12.72%, and 15.20% lower than that of the PBGC-NWF, respectively. While the porosity of MWF1, MWF2, MWF3, and MWF4 were 5.07%, 8.02%, 5.86%, and 8.53% lower than that of the PBGC-UWF. For all the data on the cumulative pore volume and the porosity, the PBGC-MWF4 value was the most notable one.

In addition, in Figure 9, it can be observed that there was a distinct boundary between the PBG matrix and the waste glass fibers, implying no chemical reaction occurs between the waste glass fibers and the PBG matrix. To further investigate whether the chemical reaction occurred between the treated fibers and the PBG matrix, XRD was applied to explore the phase structure of the final materials. As demonstrated in Figure 11, the main component of all the PBGCs was CaSO_4_·2H_2_O, indicating that the UWF or MWF had not participated in the reaction in the hydration process of gypsum. They only played the role of reinforcement bridges, providing the solid binding force, interface bonding force, and mechanical meshing force.

Figure 12 displays the flexural strength and compressive of the PBGC with the waste glass fibers and new glass fibers dosage of 1.0%. It can be seen that the strengths could meet the requirements based on the Chinese Standard of GB/T 9776-2008 test. Compared with the PBGC-NWF, the flexural strengths of the PBGC-UWF, PBGC-MWF1, PBGC-MWF2, PBGC-MWF3, and PBGC-MWF4 have enhanced by 25.47%, 38.31%, 38.57%, 38.12%, and 44.24%, respectively. Meanwhile, the compressive strengths also increased by 6.67%, 9.74%, 10.38%, 9.88%, and 13.04%, respectively. In Figure 12, these modifications of the PBGC, the PBGC-MWF4 had the best properties. The flexural and compressive strength of the PBGC-NGF presented better mechanical properties. Compared with the PBGC-NGF, the flexural and compressive strength of the PBGC-MWF4 have decreased by 4.97% and 3.87%. Taking the PBGC-UWF as a comparison, the flexural strengths of the PBGC-MWF1, PBGC-MWF2, PBGC-MWF3, and PBGC-MWF4 were increased by 10.22%, 10.43%, 10.07%, and 14.96%, respectively, and the compressive strengths raised 2.87%, 3.48%, 3.03%, and 5.96%, respectively. It can be claimed that the application of the MWFs was effective to improve the strength of the PBGC due to the relatively low porosity, high density, and strong adhesion.

To explore the effect of the MWF dosage on the mechanical properties, the flexural and compressive strengths of the PBGCs with the waste glass fibers dosages of 0.5%, 1%, 1.5%, 2%, and 2.5% were tested, as shown in Figure 13. It can be observed that the flexural and compressive strengths were reinforced with increasing fiber contents from 0.5% to 2%. Taking PBGC-MWF4 as an example, the flexural strengths were increased by 16.01%, 14.95%, 17.15%, 18.67%, and 18.14% with the fiber contents of 0.5%, 1%, 1.5%, 2%, and 2.5%, and the compressive strengths were increased by 6.58%, 5.96%, 6.29%, 11.92%, and 10.08%. The PBGC-MWF4 with 2% dosage exhibited the highest compressive strength of 9.98 MPa and flexural strength of 5.19 MPa. The trend of the PBGC-NGF mechanical properties was shown to be the same as the PBGC-MWF4. In Figure 13, the mechanical properties of the PBGC-NGF are approximately the same as the PBGC-MWF4. Figure 14 reveals the excellent performance of raised MWF4 or new glass fibers, but the dosages of 2% and 2.5% displayed that the flexural and compressive strengths had no significant differences. Figure 12 and Figure 13 proved that the WFs modified by dopamine and benzonitrile work well.

Figure 14 shows the testing process of the flexural strength failure of the PBGC under constant loading. It can be seen that the PBGC-NWF block appeared to be a complete failure under the external loading. However, due to the addition of the waste glass fibers, the failure of the block has been slightly controlled. More importantly, after the surface treatment, the PBGC-MWFs only had the fracture trace (Figure 14c–f). The weakest fracture sign came out on the PBGC-MWF4 surface (Figure 14f). Similarly, Figure 15 shows the compressive failure testing process with a constant loading [39]. The PBGC-NWF surface appeared to have more apparent longitudinal cracks and was separated into two parts (Figure 15a). At the same time, those with the addition of waste glass fibers only presented slight longitudinal or transverse cracks. These phenomena can be explained by the fact that the waste glass fibers were randomly distributed in the PBG matrix and formed a network structure with a constraint effect due to the bridge effect, which limited the crack propagation and reinforced the composites [40,41].

## 4. Conclusions

This study proposed an effective treatment method for the waste glass fibers derived from the WTB. Four modifiers involved dopamine mono case and dopamine compound with terephthalic, cinnamic, or benzonitrile have satisfactorily adhered to the surface of the waste glass fibers, demonstrating the feasibility of this approach by using terephthalic, cinnamic, or benzonitrile as substitutes for dopamine. The modification effects were exhibited by SEM/EDS, FT-IR, XPS analyses, and water contact angle measurement. It was found that the distinct roughness layer with oxhydryl or amino or nitrile groups was coated on the MWF surfaces to alter their chemical structure, resulting in outstanding hydrophilicity and reactivity. In particular, the MWF4 surface modified by dopamine compound with benzonitrile always kept a contact angle of nearly zero with outstanding hydrophilicity. The mechanical property testing results showed that the MWF improved the interfacial adhesion between the fibers and the PBC matrix, enhancing their flexural and compressive strengths. In comparison with PBGC-UWF, the compressive strength of the PBGC-MWF1, PBGC-MWF2, PBGC-MWF3, and PBGC-MWF4 increased by 2.87%, 3.48%, 3.03%, and 5.96%, and the flexural strength enhanced 10.22%, 10.43%, 10.07%, and 14.96%, respectively, with the porosity decreasing 5.07%, 8.02%, 5.86%, and 8.53%, respectively.

## Figures and Tables

**Figure 1 molecules-27-08668-f001:**
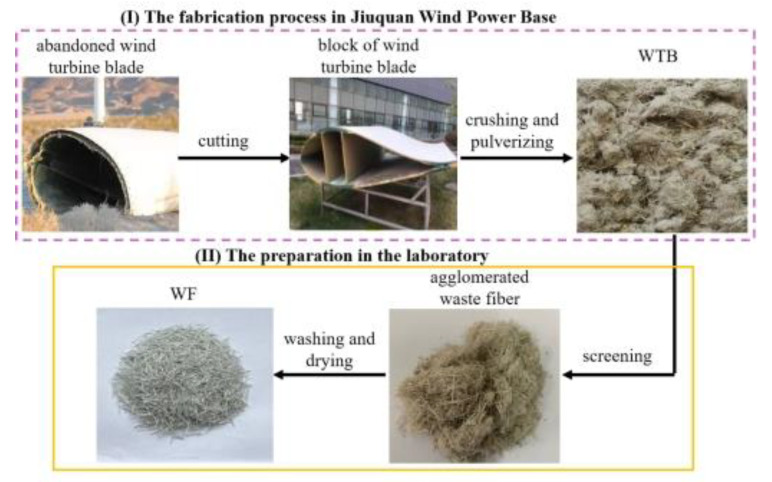
The preparation process of the waste glass fibers.

**Figure 2 molecules-27-08668-f002:**
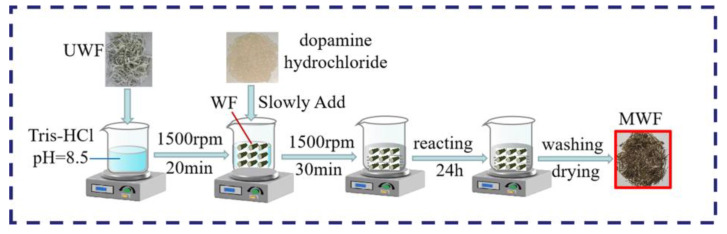
The surface modification flowchart of the waste glass fibers.

**Figure 3 molecules-27-08668-f003:**
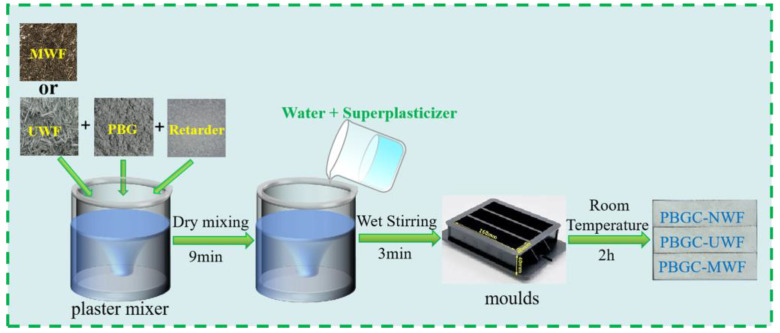
The flowchart of the PBGC preparation.

**Figure 4 molecules-27-08668-f004:**
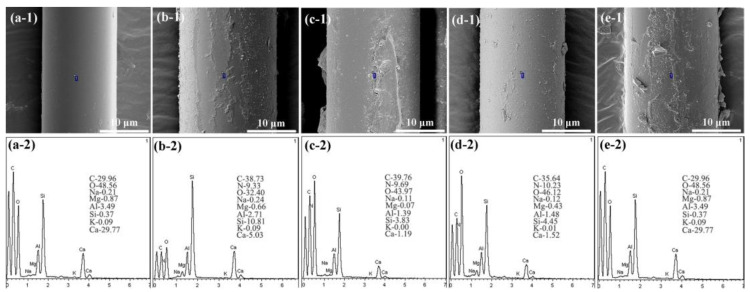
SEM morphology and EDS results of five types of waste glass fibers before and after treatment. (**a-1**) SEM image of UWF; (**a-2**) EDS image and the percentages of the elements present in the UWF; (**b-1**) SEM image of MWF1; (**b-2**) EDS image and the percentages of the elements present in the MWF1; (**c-1**) SEM image of MWF2; (**c-2**)EDS image and the percentages of the elements present in the MWF2; (**d-1**) SEM image of MWF3; (**d-2**) EDS image and the percentages of the elements present in the MWF3; (**e-1**) SEM image of MWF4; (**e-2**)EDS image and the percentages of the elements present in the MWF4.

**Figure 5 molecules-27-08668-f005:**
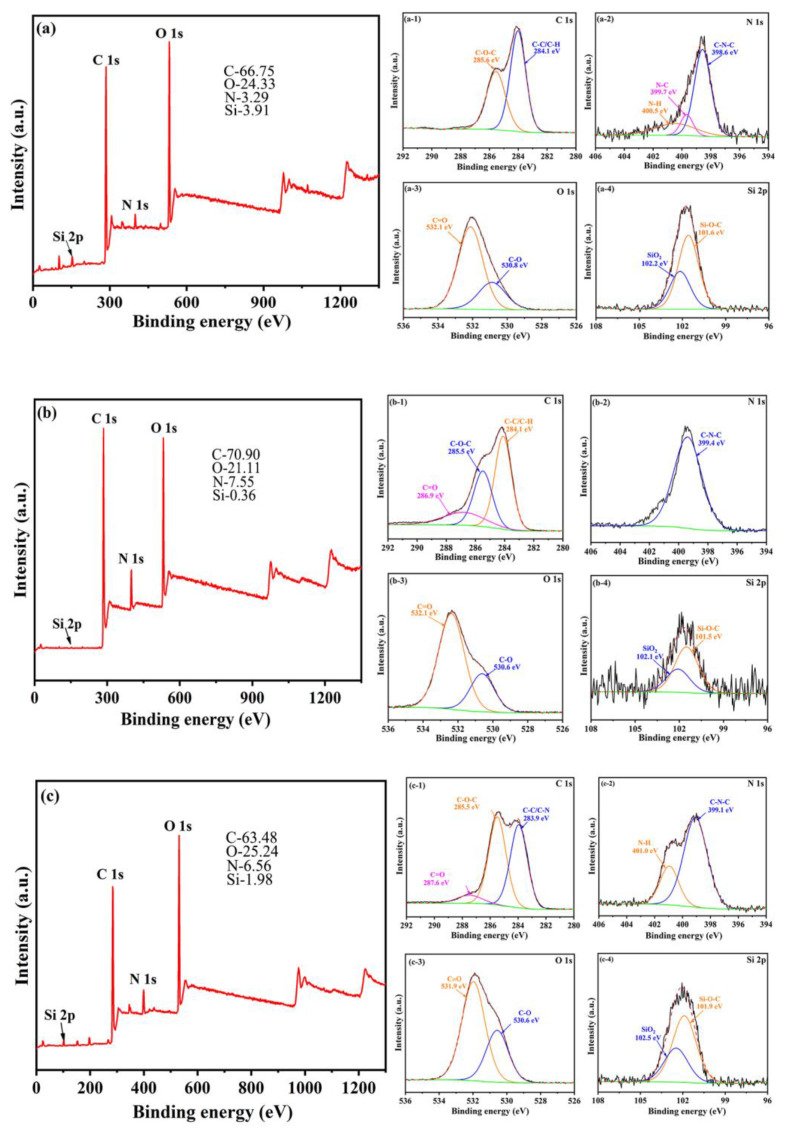
Fully scanned XPS spectra and zoomed spectra of C1s, N1s, O1s, and Si2p. (**a-1**–**a-4**) C1s, N1s, O1s, and Si2p spectra of UWF; (**b-1**–**b-4**) C1s, N1s, O1s, and Si2p spectra of MWF1; (**c-1**–**c-4**) C1s, N1s, O1s, and Si2p spectra of MWF2; (**d-1**–**d-4**) C1s, N1s, O1s, and Si2p spectra of MWF3; (**e-1**–**e-4**) C1s, N1s, O1s, and Si2p spectra of MWF4.

**Figure 6 molecules-27-08668-f006:**
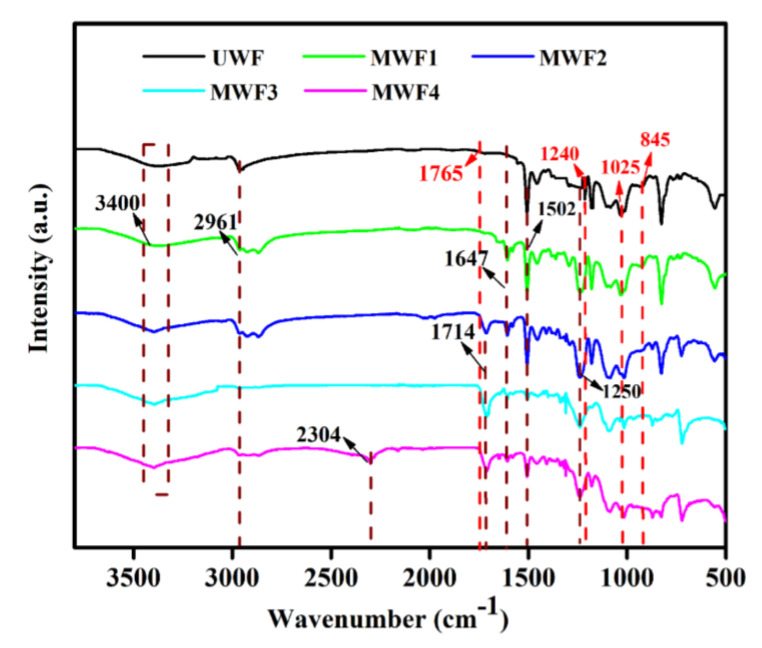
FTIR spectra of UWF and MWFs.

**Figure 7 molecules-27-08668-f007:**
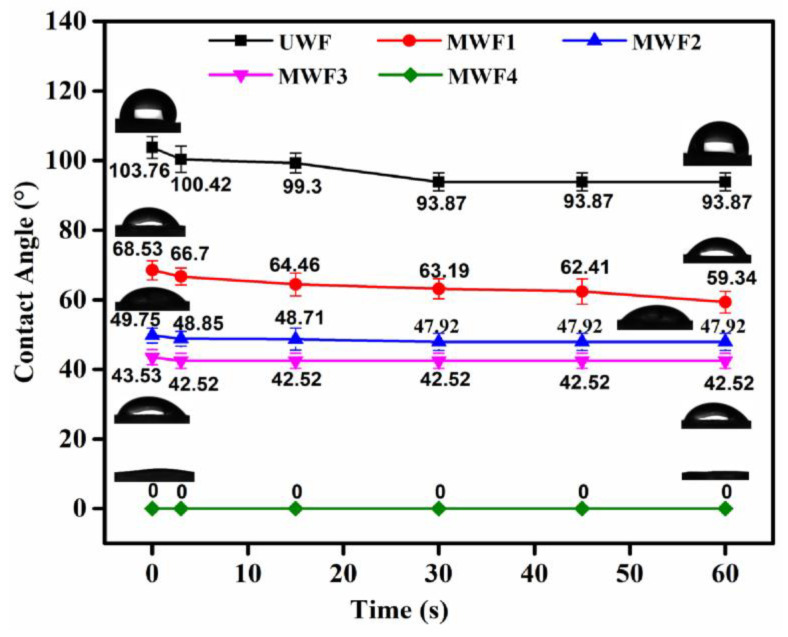
Contact angles of the UWF and MWFs.

**Figure 8 molecules-27-08668-f008:**
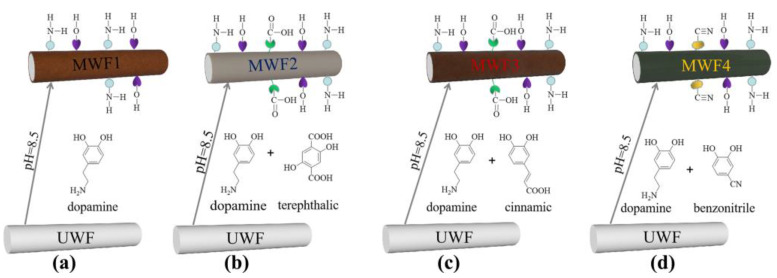
Schematic description of the coating on UWF. (**a**) dopamine, (**b**) dopamine and terephthalic, (**c**) dopamine and cinnamic, (**d**) dopamine and benzonitrile.

**Figure 9 molecules-27-08668-f009:**
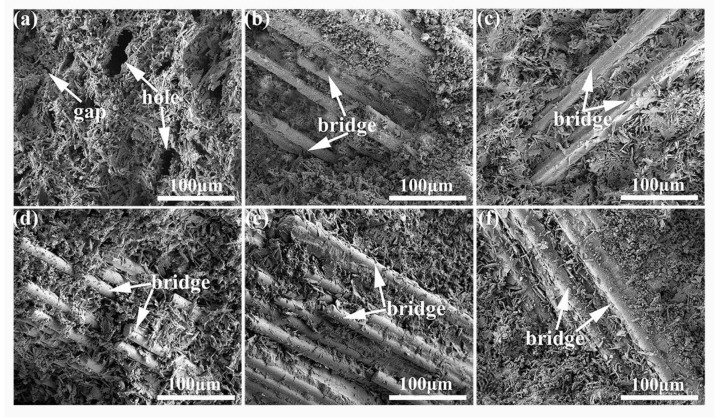
The SEM images of all the PBGC. (**a**) PBGC-NWF; (**b**) PBGC-UWF; (**c**) PBGC-MWF1; (**d**) PBGC-MWF2; (**e**) PBGC-MWF3; (**f**) PBGC-MWF4.

**Figure 10 molecules-27-08668-f010:**
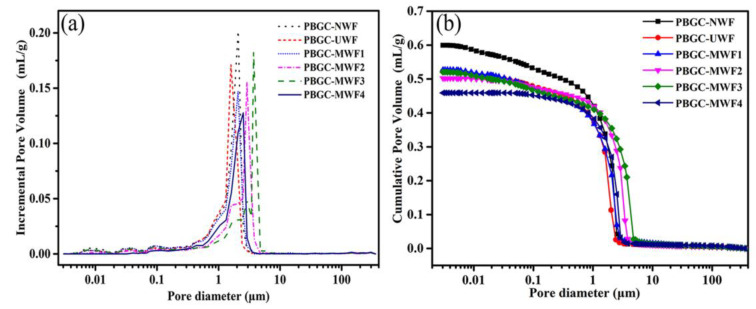
The pore size distribution and cumulative pore volume of all the PBGCs. (**a**) Pore size distribution; (**b**) cumulative pore volume.

**Figure 11 molecules-27-08668-f011:**
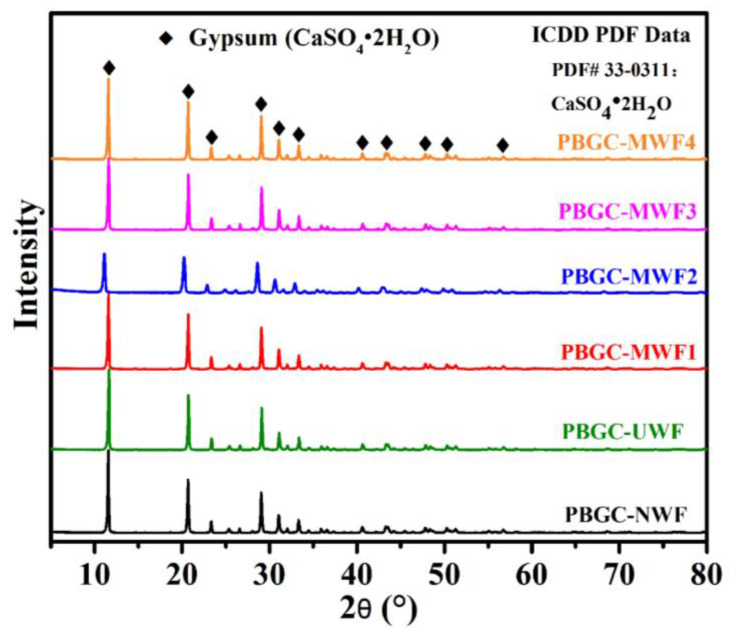
The XRD pattern of all the PBGCs.

**Figure 12 molecules-27-08668-f012:**
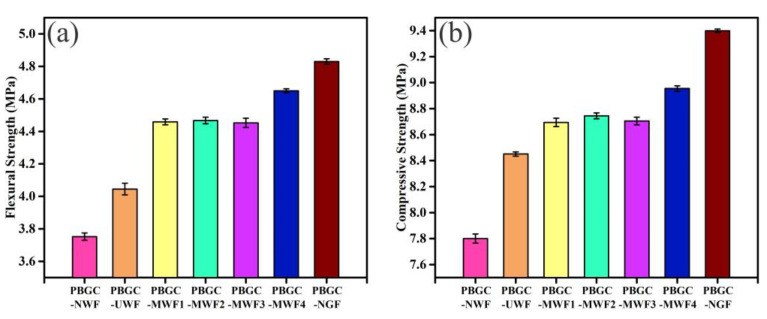
Effects of the different MWF and NGF on the mechanical properties at fixed WF contents of 1.0 %. (**a**) Flexural strength; (**b**) Compressive strength.

**Figure 13 molecules-27-08668-f013:**
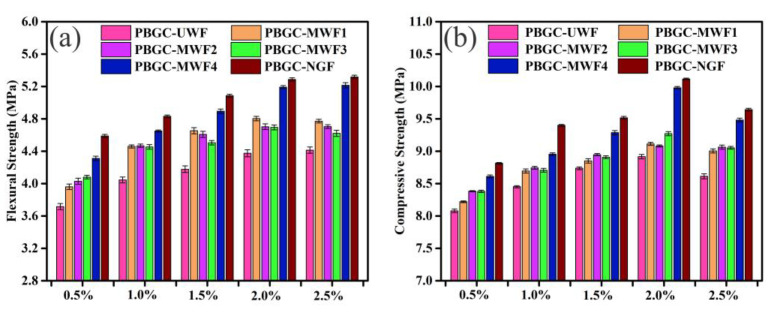
The effect of the MWF and NGF dosage on the mechanical properties. (**a**) Flexural strength; (**b**) Compressive strength.

**Figure 14 molecules-27-08668-f014:**
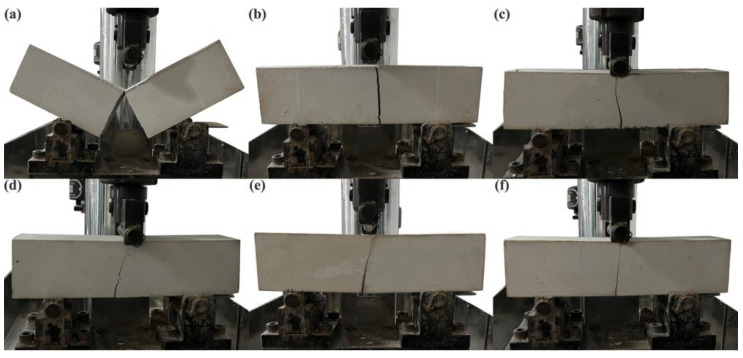
The failure testing process of the flexural strength. (**a**) PBGC-NWF; (**b**) PBGC-UWF; (**c**) PBGC-MWF1; (**d**) PBGC-MWF2; (**e**) PBGC-MWF3; (**f**) PBGC-MWF4.

**Figure 15 molecules-27-08668-f015:**
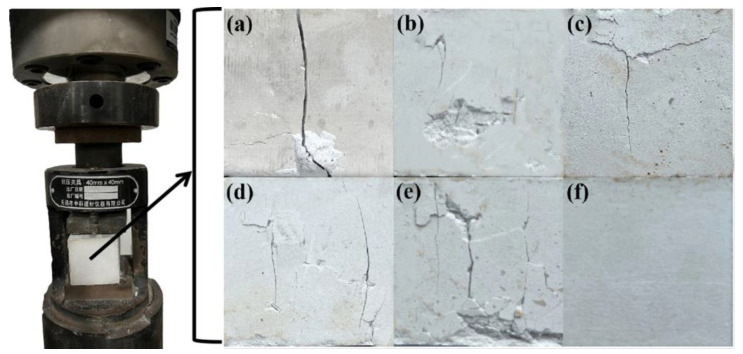
The failure testing process of the compressive strength. (**a**) PBGC-NWF; (**b**) PBGC-UWF; (**c**) PBGC-MWF1; (**d**)PBGC-MWF2; (**e**) PBGC-MWF3; (**f**) PBGC-MWF4.

**Table 1 molecules-27-08668-t001:** The chemical composition of PBG.

SO_3_	CaO	SiO_2_	Al_2_O_3_	SrO	MgO	Fe_2_O_3_	P_2_O_5_	K_2_O	CuO	ZrO_2_
54.02	38.99	2.82	0.21	0.09	0.07	0.36	1.16	0.05	68ppm	73ppm

**Table 2 molecules-27-08668-t002:** Mix proportions of PBGC samples.

Sample	Water/Plaster Ratio	PBG (g)	WF(g)	Retarder (g)	Superplasticizer (g)
NWF	0.65	1000	0	0.2	9
UWF or MWF or NGF	0.65	1000	5	0.2	9
UWF or MWF or NGF	0.65	1000	15	0.2	9
UWF or MWF or NGF	0.65	1000	20	0.2	9
UWF or MWF or NGF	0.65	1000	25	0.2	9

**Table 3 molecules-27-08668-t003:** Pore structure parameters of all the PBGC.

Sample	Median Pore Diameter (μm)	Cumulative Pore Volume (mL/g)	Porosity (%)
PBGC-NWF	1.82	0.601	58.87
PBGC-UWF	1.65	0.533	54.58
PBGC-MWF1	1.80	0.527	51.81
PBGC-MWF2	2.81	0.502	50.20
PBGC-MWF3	3.24	0.502	51.38
PBGC-MWF4	2.27	0.459	49.92

## Data Availability

Not applicable.

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
