# Peer review of "Surface-Treated Recycling Fibers from Wind Turbine Blades as Reinforcement for Waste Phosphogypsum"

_molecules, 2022, doi:10.3390/molecules27248668_

Round 1

Reviewer 1 Report

(1)What is the difference between WTB waste glass fibers and other waste glass fibers? Please clarify.

(2)L113. Why do you choose those 4 types of surface modifiers?

(3) Do wind turbine blades contain the same type of glass fibers? Please provide more details of the waste glass fibers and the process. 

(4)Please provide more details of the surface treatment of the WFs. For example, the stirring rate and the drying time.

(5) Many figures are too small. Please provide bigger and more clear figures.

(6)L131 Please provide more details of the PBG, retarder and superplasticizer.

(7) Figs. 9 and 10. Have you compared the PBGC with normal glass fibers as control samples? 

(8)Please provide more details of the compression tests.

Author Response

Reviewer #1

Thank you for your valuable comments on our article. According to your suggestions, we have extensively revised the manuscript listed as follows:

Q: What is the difference between WTB waste glass fibers and other waste glass fibers? Please clarify.

Response: We agree and have supplemented the information to clarify this issue. Figure 7 was revised to clarify the details of the raw WF and the WTB. All changes were marked in red. “The peaks at 1765 cm-1, 1240 cm-1 (aromatic C=O), and 845 cm-1 (epoxy group Antisymmetric Vibration) presented the thermoset polymer epoxy group. The peaks at 845cm-1and 770cm-1 evidenced the Si-O band, this presence of glass fiber. This abundant evidence proved WF contained glass fiber and epoxy resin.” (Please see p.8, lines 258-261). Unlike standard glass fiber, the WF of WTB was wrapped with a layer of epoxy, and these fibers were not pure glass fibers.

Q: L113. Why do you choose those 4 types of surface modifiers?

Response: In previous modification methods, such as the chemical modification or adding various admixtures, the bonding strength of the interface between the waste fiber and gypsum matrix is still challenging due to the defects of the interfaces. It was found that the wettability of the fiber surface is critical to decreasing the defects of the fiber surfaces. Therefore, the authors used these four modifiers to change the hydrophilic character of the waste fibers and reduce the defects of the interfaces. After a coated hydrophilic film on the surface of the waste fiber, the mechanical properties of the composites can be considerably improved.

Q: Do wind turbine blades contain the same type of glass fibers? Please provide more details of the waste glass fibers and the process.

Response: As suggested by the reviewer, a short paragraph has been added in the Introduction section: “Recently, wind turbine blade composites contained glass fiber and epoxy. In the wind turbine blade manufacturing process, several high-strength fibers, such as carbon fibers, basalt, aramid fibers, S-glass, R-glass, and E-glass fibers have been used as reinforcement agents. However, considering the cost factor, in China, wind turbine blades usually use E-glass fiber as the reinforced materials for economic and environmental reasons.” (Please see p.1, lines 40-48).

We added a short paragraph in part 2.2 to describe the waste glass fibers: “which peer blocks about 1 m2. Then, shred the blocks crushed by a crusher and pulverize the fibrous with a grinding mill. (The factory specially customized the machine of the crusher and grinding mill. The equipment had four parts: shredding part, transmission equipment, primary crushing equipment, and vibrating screen). The high-performance multi-functional grinding mill (XICHU-4500Y, ZheJiang, China) ground the sheets into fibrous. The fibrous materials were agglomerated like cotton. The second step was sieving, washing with deionized water to remove the floc, and drying the fibers in an oven at 80 oC.” (Please see p.3, lines 120-127).

Q: Please provide more details of the surface treatment of the WFs. For example, the stirring rate and the drying time.

Response: The authors have added the details in part 2.3 to describe the surface treatment of the WFs: “As illustrated in this figure, first, 25g of UWFs were immersed in 250mL Tris-HCl buffer solution with pH=8.5 and stirred for 20 min, with a stirring rate of 300 r/ min, to obtain the UWF suspension at room temperature [32]. Subsequently, 0.5g modifiers were divided into five parts and added into the above suspension every five minutes, and obtained a uniform solution.” (Please see p.4, lines 139-142).

“Finally, the products were collected by filtration and dried at 80 oC for 2h to obtain the MWF1.” (Please see p.4, line 146).

Q: Many figures are too small. Please provide bigger and more clear figures.

Response: The authors have carefully checked and corrected these issues in the revised manuscript. Most of the figure numbers and images have been revised. (Please see figure 1-16). Especially Figure 6 and Figure 8 (Please see p.7-8, lines 246-250; p.9, line 285), the images of the Contact Angle have been added. The difference between the four modification methods can be directly observed.

Q: L131. Please provide more details of the PBG, retarder and superplasticizer.

Response: In figure 3, the cementitious material is (PBG). “An effective way is that PG is pretreated and calcined to manufacture the phosphorus-building gypsum (PBG) products.” (Please see p.2, lines 63-65).

“In this paper, according to the phosphogypsum-water ratio of 1:2.5, washed the phosphogypsum until it was neutral. Some organic pollutants, soluble phosphorus, fluorine, and alkali could be removed. Then calcined in an oven at 180 oC for 3h, and ground with a ball mill followed by15 days curing to obtain PBG.” (Please see p.4, lines 160-163).

We added the information of retarder and superplasticizer. “In this paper, the retarder was tartaric acid, and the superplasticizer was a polycarboxylic water-reducer.” (Please see p.4, lines 170-171).

Q: Figs. 9 and 10. Have you compared the PBGC with normal glass fibers as control samples?

Response: The authors have changed Figures 9 and 10 in the revised manuscript.

In section 2.4, we defined the PBGC-NGF as “The length of the new glass fiber was about 9mm, similar to the WF, the PBGC with new glass fibers was named PBGC-NGF.” (Please see p.4, lines 168-169; p.4, lines 171; p.12, lines 351).

Table 2 changed the contents of the Sample. (Please see p.5, lines 185).

A short paragraph “In Figure 13, these modifications of the PBGC, the PBGC-MWF4 had the best properties. The flexural and compressive strength of the PBGC-NGF presented better mechanical properties. Compared with the PBGC-NGF, the flexural and compressive strength of the PBGC-MWF4 have decreased by 4.97% and 3.87%.” has been added in the revised manuscript. (Please see p.12, lines 356-360).

“In figure 14, the mechanical properties of the PBGC-NGF are approximately the same as the PBGC-MWF4. Figure 14 reveals the excellent performance of raised MWF4 or new glass fibers, but the dosages of 2% and 2.5% displayed that the flexural and compressive strengths had no significant differences. Figures 13 and 14 proved that the WFs modified by dopamine and benzonitrile work well.” (Please see p.13, lines 377-382).

Q: Please provide more details of the compression tests.

Response: We added the details of the compression tests. “The loading rate of the flexural and compressive strengths was 50 N/s and 2mm/min. First, test the flexural strength of three 40mm × 40mm × 160 mm specimens, and each of the half specimens after flexural strength testing were used as the compressive strength testing.” (Please see p.5, lines 189-193).

Reviewer 2 Report

Regarding the paper entitled “Surface-treated recycling fibers from wind turbine blades as  reinforcement for waste phosphogypsum”, authors tried to use four modifiers to modify the waste fiber to form a better reaction with phosphogypsum. The title is interesting and the paper contains enough data to support the title. The introduction section is well-written and understandable. Totally, the paper can be published after minor corrections suggested below:

It has been stated that each experiment was repeated 3 times. Pl explain about the error in the results in a given level of confidence.

Part 2.1. (Materials): Pl mention the composition of phosphogypsum.

Part 2.2. The waste glass fibers: Pl explain more about the Crushing and Grinding stages. What kind of crusher-Mill was used? What was the final size of waste glass fibers?

2.3. Surface treatment of the WFs: Pl re-check the mass of modifiers used. 0.5 g or 0.25 g?

Figure 4: Pl explain more about the relationships between the contact angle and the elemental compositions of the surfaces.

Figure 8: What does it mean by PDF33-0311?

Author Response

Reviewer #2

Regarding the paper entitled “Surface-treated recycling fibers from wind turbine blades as reinforcement for waste phosphogypsum”, authors tried to use four modifiers to modify the waste fiber to form a better reaction with phosphogypsum. The title is interesting and the paper contains enough data to support the title. The introduction section is well-written and understandable. Totally, the paper can be published after minor corrections suggested below:

Response: We sincerely appreciate your professional review work on our work. We have extensively revised our previous draft according to your suggestions, and the detailed corrections are listed below.

Q: It has been stated that each experiment was repeated 3 times. Pl explains about the error in the results in a given level of confidence.

Response: There might be two reasons for the errors in this paper. One reason was that each group of samples was composed of three specimens, and the model itself had a slight difference in the model preparation. The test data showed the average value of the three flexural and six compressive strengths. There would be errors between each sample and the average value. Another reason is that the fibers were not uniformly distributed throughout the cross-section and longitudinal sections in the gypsum matrix. Analysis showed the resulting error bars.

Q: Part 2.1. (Materials): Pl mentions the composition of phosphogypsum.

Response: The authors have added the details of the composition of phosphogypsum. “Table 1 lists the composition of the PBG by the X-ray fluorescence (Tiger S8, BRUKER, Germany).” (Please see p.4, lines 164-165).

Table 1. The chemical composition of phosphogypsum

SO3

CaO

SiO2

Al2O3

SrO

MgO

Fe2O3

P2O5

K2O

CuO

ZrO2

54.02

38.99

2.82

0.21

0.09

0.07

0.36

1.16

0.05

68ppm

73ppm

(Please see p.5, lines 184).

Q: Part 2.2. The waste glass fibers: Pl explain more about the Crushing and Grinding stages. What kind of crusher-Mill was used? What was the final size of waste glass fibers?

Response: A short paragraph has been added to the revised manuscript. “In this step, the abandoned WTB were cut into blocks using a diamond wire saw, which peer blocks about 1 m2. Then, shred the blocks crushed by a crusher and pulverize the fibrous with a grinding mill. (The factory specially customized the machine of the crusher and grinding mill. The equipment had four parts: shredding part, transmission equipment, primary crushing equipment, and vibrating screen.). The high-performance multi-functional grinding mill (XICHU-4500Y, Zhejiang, China) ground the sheets into fibrous. The fibrous materials were agglomerated like cotton. The second step was sieving, washing with deionized water to remove the floc, and drying the fibers in an oven at 80 oC. The final products were the WF with a length of about 4-9mm, and the length-diameter ratio was not unique.” (Please see p.3, lines 120-128).

Q: 2.3. Surface treatment of the WFs: Pl re-check the mass of modifiers used. 0.5 g or 0.25 g?

Response: We have carefully checked and revised this part as “Subsequently, 0.5g modifiers were divided into five parts and added to the above suspension every five minutes, finally for a uniform solution.” (Please see p.4, lines 142-143). “The other three modification steps were similar to the MWF1, which used 0.25g dopamine with 0.25g terephthalic, cinnamic, or benzonitrile, respectively, to react for 24h at room temperature could acquire the MWF2, MWF3, and MWF4.” (Please see p.4, lines 147-149).

Q: Figure 4: Pl explains more about the relationships between the contact angle and the elemental compositions of the surfaces.

Response: It can be seen that the initial contact angle of the waste fiber was 103.76°, which confirmed the hydrophobic innate of the surface. After the chemical treatment of dopamine and dopamine with terephthalic, cinnamic, or benzonitrile, the contact angle of the waste fibers declined due to the surface containing hydrophilicity groups. EDS presented little N element on the surface of an unmodified fiber. Four modifications altered the content of the N element, which was significantly increased. Amino groups were derived from dopamine. In the literature, researchers found that polydopamine has a function that could firmly adhere to various substrates, including plastics and glasses. (Dreyer et al., 2012) Furthermore, FTIR and XPS illustrated a series of diffraction peaks belonging to hydroxyl, amino, and nitrile that appeared on the fibers. Therefore, the contact angle of the fiber surfaces from 103.76° to disappears.

Q: Figure 8: What does it mean by PDF33-0311?

Response: Thanks very much for pointing out this question. We have carefully checked and corrected the mistake. In Figure 12, we added the ICDD PDF Data; “PDF33-0311” meant the CaSO4·2H2O. (Please see p.12, lines 348).

Reviewer 3 Report

1. What is the innovation of the manuscript compared with the previous researches? Just change the raw material-Phosphogypsum.

2. PG, not PBG is the short name for phosphogypsum.

3. From figure 3, PG is directly used as raw material to make PBGC. There is no cementitious material. Why the PBGC samples have good mechanical property?

4. There are many soluble components, such as P, F, in PG. These soluble components tend to cause frost in the produced materials. How to solve the problem in the manuscript?

Author Response

Reviewer #1

Response: Thanks for the reviewer’s valuable comments and suggestions. We modified our manuscript according to these suggestions, each of which was briefly described below.

Q: What is the innovation of the manuscript compared with the previous researches? Just change the raw material-Phosphogypsum.

Response: The authors have added the contribution of this work compared with previous researches. A short paragraph “Unlike previous studies, in which new glass fibers were mixed with gypsum matrix, the mechanical properties of the waste fiber/gypsum composites will be declined sharply due to the poor bonding strength between the waste glass fibers and the gypsum matrix” has been added in the revised manuscript (Please see p.2, lines 83-87).

Q: PG, not PBG is the short name for phosphogypsum.

Response: We have revised this part according to this comment. (Please see p.2, line 61-63; p.2, lines 53-55).

Q: From figure 3, PG is directly used as raw material to make PBGC. There is no cementitious material. Why the PBGC samples have good mechanical property?

Response: The authors have added the definition of the PBG in the Introduction: “An effective way is that PG is pretreated and calcined to be prepared all kinds of phosphorus-building gypsum (PBG) products.” (Please see p.2, lines 63-65).

In figure 3 the cementitious material was PBG and had much better mechanical properties. Use the PBG, UWF or MWF, or NGF to prepare the PBGC-NWF, PBGC-UWF, PBGC-MWF, and PBGC-NGF. It has been revised as “The length of the new glass fiber was about 9mm, similar to the WF, the PBGC with new glass fibers was named PBGC-NGF”. (Please see p.5, lines 168-169).

Q: There are many soluble components, such as P, F, in PG. These soluble components tend to cause frost in the produced materials. How to solve the problem in the manuscript?

Response: We have added the details of the pretreatment of the PG and the composition of PBG. A short paragraph “In this study, with a phosphogypsum to water ratio of 1:2.5, the phosphogypsum has been washed to neutral, and the organic pollutants, soluble phosphorus, fluorine, and alkali have been removed by this washing process. After calcined in an oven at 180 oC for 3h, ground with a ball milling procedure, followed by15 days of natural-drying to obtain the PBG. Table 1 lists the composition of the PBG by the X-ray fluorescence (Tiger S8, BRUKER, Germany)” which has been added in the revised manuscript (Please see p.4, lines 160-165).

Table 1. The chemical composition of PBG.

SO3

CaO

SiO2

Al2O3

SrO

MgO

Fe2O3

P2O5

K2O

CuO

ZrO2

54.02

38.99

2.82

0.21

0.09

0.07

0.36

1.16

0.05

68ppm

73ppm

(Please see p.5, lines 184).

Round 2

Reviewer 3 Report

Most of the comments issued after the first review have been addressed.